# Peptide-Based Nanoparticles for Therapeutic Nucleic Acid Delivery

**DOI:** 10.3390/biomedicines9050583

**Published:** 2021-05-20

**Authors:** Prisca Boisguérin, Karidia Konate, Emilie Josse, Eric Vivès, Sébastien Deshayes

**Affiliations:** PhyMedExp, University of Montpellier, INSERM U1046, CNRS UMR 9214, 34295 Montpellier, France; prisca.boisguerin@inserm.fr (P.B.); karidia.konate@inserm.fr (K.K.); emilie.josse@inserm.fr (E.J.); eric.vives@umontpellier.fr (E.V.)

**Keywords:** cell-penetrating peptide, nanoparticle, nucleic acid, delivery, self-assembly

## Abstract

Gene therapy offers the possibility to skip, repair, or silence faulty genes or to stimulate the immune system to fight against disease by delivering therapeutic nucleic acids (NAs) to a patient. Compared to other drugs or protein treatments, NA-based therapies have the advantage of being a more universal approach to designing therapies because of the versatility of NA design. NAs (siRNA, pDNA, or mRNA) have great potential for therapeutic applications for an immense number of indications. However, the delivery of these exogenous NAs is still challenging and requires a specific delivery system. In this context, beside other non-viral vectors, cell-penetrating peptides (CPPs) gain more and more interest as delivery systems by forming a variety of nanocomplexes depending on the formulation conditions and the properties of the used CPPs/NAs. In this review, we attempt to cover the most important biophysical and biological aspects of non-viral peptide-based nanoparticles (PBNs) for therapeutic nucleic acid formulations as a delivery system. The most relevant peptides or peptide families forming PBNs in the presence of NAs described since 2015 will be presented. All these PBNs able to deliver NAs in vitro and in vivo have common features, which are characterized by defined formulation conditions in order to obtain PBNs from 60 nm to 150 nm with a homogeneous dispersity (PdI lower than 0.3) and a positive charge between +10 mV and +40 mV.

## 1. Introduction

Since 2016, we have observed an acceleration in the development of nucleic acid as therapeutics with the approval of several molecules by the U.S. Food and Drug Administration (FDA). For example, two therapeutics based on RNA interference (RNAi) were approved, ONPATTRO^®^ (Partisiran) for polyneuropathy in hereditary transthyretin-mediated (hATTR) amyloidosis in 2018 [1], and GIVLAARI™ (Givosiran) for acute hepatic porphyria (AHP) in 2019 [2]. More recently, resulting from the worldwide COVID-19 pandemic, two vaccines based on mRNA technology were put on the market by the companies Pfizer/BioNTec [3] and Moderna [4].

Oligonucleotides (ONs) are short polymers of nucleic acids (RNAs or DNAs), which could be natural or chemically modified. The use of therapeutic ONs to treat a wide range of diseases has expanded the range of possible targets beyond what is generally accessible by conventional pharmaceutics, such as gene silencing, splice modulation, or gene activation. Several examples could be mentioned, such as the antisense oligonucleotides (ASOs, 15 to 20 nucleotides), acting primarily in the nucleus by selectively cleaving pre-mRNAs having complementary sites via an RNase H dependent mechanism [5]. Subsequently, double-stranded short interfering RNAs (siRNAs) that contain 20–25 nucleotides were developed as major therapeutic tools for silencing gene expression. The double-stranded siRNA is separated by helicase, and the antisense strand (or guide strand) is embedded into the RNA-induced silencing complex (RISC) to guide it to the complementary target mRNA for degradation [6,7]. Micro RNAs (miRs) are an endogenous, highly conserved, small non-coding RNA composed of 20–24 nucleotides that have been implicated as key regulators of target gene expression [8]. At the post-transcriptional level, miRs bind to the 3′-untranslated regions of the corresponding target mRNAs of protein-coding genes, thereby resulting in target mRNA degradation and the inhibition of mRNA translation.

Longer nucleic acids (NAs) such as therapeutic DNAs are mainly used in the form of plasmids (pDNA), which encode specific genes or regulatory sequences for endogenous proteins [9]. In this context, the suppressor gene p53 is the most widely transferred gene in clinical trials due to the fact that it is one of the most frequently mutated genes in different types of cancer [10]. In 2003, the Chinese company Shenzhen SiBiono GenTech obtained approval from the State Food and Drug Administration of China for its recombinant adenovirus-based p53 gene therapy for head and neck squamous cell carcinoma [11]. Another example is the most recently authorized gene therapy drug by the U.S. Food and Drug Administration (May 2019) called Zolgensma (Novartis) for the expression of the survival motor neuron 1 protein (SMN1) in motor neurons for the treatment of spinal muscular atrophy (SMA) [12]. pDNA can also be used to edit genes via the CRISPR mechanism through the internalization of plasmids coding the Cas9 protein and the RNA guide strand with the targeted gene sequence.

Despite significant advances in different therapeutic NA applications, a major obstacle preventing their widespread usage is the challenge of organ- and tissue-specific delivery. To overcome this bottleneck, several strategies have been employed such as the chemical modification of the nucleic acid to improve its ‘drug-likeness’, as well as the use of cell-targeting or cell-penetrating moieties for covalent conjugation or nanoparticle formulation. More than twenty years ago, cell-penetrating peptides (CPPs) were identified as potential carriers for a wide variety of biomolecules, including NAs [13,14,15]. Usually defined as short (up to 30 amino acids) peptides that originate from different sources (e.g., humans, mice, viruses or purely synthetic), CPPs were developed as one of the most promising non-viral strategies for improving the intracellular routing of NAs, since they constitute a great alternative to the existing viral (adenoviruses, retrovirus, etc.), lipid-based, or polymer-based methods [16].

Initially applied through covalent conjugation to NAs, CPPs were increasingly used in non-covalent strategies based on electrostatic and hydrophobic interactions between both CPPs and NAs (Figure 1). These interactions resulted in the self-assembly of peptides with NAs and the formation of peptide-based nanoparticles (PBNs), thus opening peptides to the field of nanomedicine [17,18,19]. The delivery of NAs by peptides has become a separated subfield in the research domain of CPPs due to the formation of larger intermolecular structures instead of the monomolecular solutions of covalent CPP-NA conjugates. Moreover, peptide-based vectors are now considered to be suitable candidates for the delivery of therapeutic NAs due to their easy automated synthesis, single-step formulation, and biocompatible properties.

Among the large number of CPPs, several amphipathic peptides were designed with both hydrophilic and hydrophobic domains in order to confer both NA complexation and membrane interaction abilities. Primary amphipathic CPPs have these two domains distributed according to each amino acid position along the peptide chain, as shown for Mgpe and MPG [20,21], while secondary amphipathic peptides result from the formation of both hydrophilic and hydrophobic domains through the secondary structure folding [22,23]. Many of the CPPs of this later class were used to form PBNs such as PepFect [24], RICK [25], or WRAP [26]. However, although originally based on amphipathic peptides, the nanoparticles approach has been extended to all peptides and peptide analogues that were able to form stable CPP:NA nanoparticles and to improve NA delivery into mammalian cells [19,27]. The ability of CPPs to form PBNs has been associated with several structural properties. While the conformational state of non-covalent CPPs has been shown to play an important role in the interaction with NAs, as well as in the self-assembly process leading to efficient PBNs [26,28], the role of physicochemical parameters such as the amphipathicity, charges, and presence of specific residues is directly related to PBN efficiency [21].

In this review, we present different peptide families used for PBN formulation in the presence of various NAs (Table 1), which were used and published during the period between 2015 and 2021 (even if their design was reported years previously). In particular, we have focused this review on peptide-based nanoparticles formed by the self-assembly of peptides, which are essentially native or modified CPPs (PEGylated, grafted to fatty acids or fusogenic moieties, etc.). CPPs used in the covalent strategy or to decorate other nanoparticles such as polymers, nanotubes, or even viruses are reported elsewhere [29,30].

In detail, we have first summarized seven “main” CPP families: the poly-cationic, GALA/KALA/RALA, PepFect/NickFect, CADY-K/RICK, WRAP, C6, and Mpge families, and secondly highlighted new developed CPPs with high potential (see Section 9 “Other CPPs forming PBNs”). Finally, we have recapitulated known PBN optimization methods such as PEGylation and different targeting strategies which are important for the development of “intelligent” PBNs in view of pharmacological applications.

## 2. Poly-Cationic Family

Peptides made only with lysine and arginine, named poly-lysine (Poly-Lys) and poly-arginine (Poly-Arg), are some of the first artificial CPPs that were evaluated for their internalization in living cells. Prof. S. Futaki’s group was one of the pioneers working on the effect of the positive charge on cell transfection poly-cationic peptides with 4 to 16 residues [68]. In order to enable the nanoparticle formulation, Poly-Arg sequences grafted with different fatty acids were compared, and the stearylation of R8 was shown to improve pDNA transfection. This finding was surprising because StA-R8:pDNA particles were larger than those formulated with naked R8 or with LipofectAMINE [69]. Additionally, for siRNA delivery, long chain fatty acids such as oleic acid or stearic acid grafted on R8 could efficiently improve nanoparticle formation, resulting in higher survivin silencing in cancer cells, to cite one example [31].

Other approaches were developed by Alhakamy and colleagues using Poly-Lys- [32] and Poly-Arg-based [33] nanoparticles in the presence of Ca2+ ions. In both cases, the presence of Ca2+ ions during formulation induced a reduction in the nanoparticles’ mean size, resulting in an increased pDNA cellular transfection in different cell lines.

A histidine-modified arginine-rich CPP (HR9) was able to form non-covalent stable complexes, with plasmid DNA encoding for the non-structural protein 3 (NS3) hepatitis C virus (HCV) [70]. As NS3 protein is known to be involved in both CD4+ and CD8+ T cells with viral clearance, its enhanced expression into HEK-293T cells compared to other transfection reagents could be an important step towards the development of an HCV gene-based vaccine.

Nowadays, Poly-Lys or Poly-Arg peptides are mainly used as grafting sequences on other nanoparticles such as liposomes, polymers, nanogold, or viral particles to increase their cellular internalization in the same way as Tat CPP [71,72,73].

## 3. GALA/KALA/RALA Family

The 30 amino acid long, amphipathic, and α-helical GALA peptide [74] was first designed as a lipid bilayer interactor at low pH due to its fusogenic properties [75]. The bilayer destabilizing properties of GALA were used to promote gene delivery in vitro in combination with poly-lysine-conjugated ligands [76]. To favor endosomal escape, GALA peptide was modified by replacing glutamate residues with lysine residues, resulting in the KALA peptide, which was also able to condense nucleic acids to nanoparticles [77]. When environmental pH decreased from 7.5 to 5.0, KALA peptides undergo a pH-dependent amphipathic α-helix to random coil conformational change, leading to entrapped cargo release. KALA has also been used in combination with poly-lysine [78], polyethylenimine (PEI) [79], and (poly (DMAEMA-NVP))-b-PEG-galactose [80] for DNA gene delivery. More recently, Katayama and co-workers found that a liposome modified with the KALA peptide was the most effective drug delivery system for mitochondrial targeting in C2C12 cells [81].

Afterwards, the KALA peptide was further modified to improve transfection efficiency by changing the lysine residues to arginine residues [34]. With seven arginine residues, the RALA peptide formed nanoparticles in the presence of anionic entities such as plasmids in a highly tunable way depending on the used molar peptide/DNA ratios, changing the size and surface charge of the PBNs. RALA PBNs were internalized via the clathrin- and caveolin-mediated endocytosis pathways, but a pH drop in the endosomes induced an increasing α-helicity of RALA, provoking the endosomal release of the transfected cargo. Modification of the RALA sequence in terms of amino acid composition and sequence length failed to improve the functional characteristics of RALA, confirming its superior sequence for non-toxic gene delivery [82]. RALA is a widely used peptide-based delivery system, mainly optimized for the transfection of different oligonucleotides such as plasmids [34,83,84], siRNA [35,36,37], mRNA [38], and for DNA vaccination [85], demonstrating its broad utility.

## 4. PepFect/NickFect Family

Based on the short transportan-derived peptide TP10, Prof. Ü. Langel’s group developed a subset of different CPPs for NA delivery. In brief, TP10 was modified with stearic acid to improve non-covalent ON-complex formation and to enhance peptide-membrane interactions [86]. This peptide, which was later named PepFect 3 (PF3), was used as a base for further modifications resulting in the widely studied analogues PepFect 6 (PF6) and PepFect 14 (PF14). The PF6 peptide was modified with endosomolytic trifluoromethylquinoline moiety, aiming to increase the endosomal escape of the peptide [24]. The PF14 peptide was designed with non-encoded ornithine residues for increased stability and improved uptake [39]. All PF complexes were described as being taken up via receptor-mediated endocytosis involving class-A scavenger receptors (SCARAs) [87,88].

More recently, based on physicochemical features in the complex formation and on the biological efficacy, a series of PF14 modifications were developed with altered charges and fatty acid contents. Kurrikoff and colleagues showed that with an optimal combination of overall charge and hydrophobicity in the peptide backbone, in vivo gene delivery can be enhanced [42]. Interestingly, Gestin and co-workers found that through an optimized high-throughput luciferase assay, small molecule drugs (MPEP, VU0357121 and Ciproxifan) induced an increased transfection efficacy of PF14 complexed to splice-correcting oligonucleotides [41]. This finding was quite surprising because it was not really clear whether the drugs influenced nanoparticle formation, and the underlying mechanism of cellular entry was not defined.

With regard to mRNA transfection, Prof. R. Brock’s group published the use of PepFect14 to formulate CPP-mRNA nanoparticles, showing efficient reporter protein expression in two- and three-dimensional cancer cell cultures [40]. More importantly, following an intraperitoneal injection of PBNs encapsulating mCherry coding mRNA, they could reveal an important mCherry protein expression within the tumors of the treated mice. This protein expression was not observed in mice treated with the naked mRNA or with the mRNA transfected with Lipofectamine MessengerMax.

More or less in parallel to the PepFect family, Prof. Ü Langel’s group developed the NickFect family from the PF3 sequence [44]. First, in order to enhance cellular uptake and endosomal release, the PF3 peptide was modified at Lys7, located within the linker between the galanin and the mastoparan residues (from the former Transportan or TP10 peptides) [43]. In detail, by replacing Lys7 with ornithine and continuing the synthesis by coupling Gly6 to the δ-NH2 group of ornithine, the authors obtained the NickFect 51 (NF51) peptide-forming PBNs in the presence of pDNA, and were able to transfect different cell types. Based on this sequence, a novel amphipathic α-helical peptide, NF55, was designed for efficient in vivo DNA delivery by modifying the net charge and the helicity of the CPP [44,45]. More recently, Freimann and co-workers presented a new formulation approach called cryo-concentration for obtaining stable and homogeneous nanoparticles showing significantly higher bioactivity in vivo [89].

## 5. CADY-K/RICK Family

Among amphipathic peptides, CADY-K [28] and RICK [25] peptides were directly derived from the secondary amphipathic 20-residues CADY peptide, which was specially developed for siRNA delivery [22]. Investigations of different CADY analogues with substitutions and mutations allowed for the optimization of the sequence and the observation that the siRNA-loaded nanoparticles formed by the CADY-K peptide, a shortened version of CADY, displayed a twofold higher biological activity than the parental peptide or other analogues [28]. CADY-K was an ideal candidate for further applications, particularly with regard to ex vivo or in vivo siRNA delivery. However, the in vivo application of CPPs could be compromised by degradation phenomena resulting from extracellular and/or intracellular proteases, probably partly explaining the low success of CPP development in clinical trials. Therefore, to overcome protease digestion, nanoparticles were formulated with a retro-inverso analogue of CADY-K, called RICK [25]. The retro-inverso transformation, meaning the synthesis of peptides with D-amino acids in the reverse sequence of the naturally occurring L-isoforms, has commonly been employed as a strategy for the development of proteolytically stable analogues maintaining both their structural features and activities [90,91,92]. Bearing a high degree of topochemical equivalence to its L-parental homologue, RICK conserved the main biophysical features of an amphipathic CPP, kept the ability to associate with siRNA in stable PBNs, and induced the knock-down of protein expression.

Interestingly, Chen and colleagues recently used the CADY peptide for the transfection of antisense oligonucleotides (ASO) targeting the acyl carrier protein (acpP) of multi-drug resistant (MDR) *Acinetobachter baumannii* (*A. baumannii*) [93]. The authors claimed that CADY:ASO NPs provided a patent strategy for the treatment of MDR-bacteria, because CADY-NPs decreased the expression of acpP in a concentration-dependent manner, resulting in a MDR-*A. baumannii* growth retardation.

## 6. WRAP Family

Studies on CADY-K and RICK peptides have emphasized the requirement for several structural properties for both PBN formation and the resulting biological activity. As already observed for most amphipathic peptides, the existence of distinct hydrophobic and hydrophilic domains was required for cargo interactions, as well as for nanoparticle formation. In addition, the analysis of amino acid composition revealed a strong redundancy of arginine and tryptophan residues [49,94,95,96]. Based on this knowledge, a new family of CPPs was conceived: WRAP (W- and R- rich amphipathic peptides) were composed of only three amino acids (leucine, arginine, and tryptophan) [26]. These short (15/16mer) peptides were able to form stable PBNs, enroll siRNA in different cell lines (U87, MCF7, Neuro2a, HT29, etc.), and trigger more than 50% luciferase silencing at low siRNA concentrations (20–50 nM, depending on the cell line). This knock-down efficiency resulted from a rapid PBN internalization within 5–15 min of incubation.

Later on, the rapid internalization of the WRAP-PBNs was associated with their internalization mechanism [97]. By combining the whole panel of available approaches, including biophysical (leakage assay), biological (dynamin triple-KO cells), confocal (endocytosis and vesicle markers), and electron microscopy experiments, our laboratory could highlight that the balance between direct translocation and endocytosis-dependent internalization clearly shifted in favor of direct translocation through the plasma membrane. Furthermore, we deduced that the low percentage of endocytosis was mainly due to naturally occurring endocytosis processes at the surface of the cells. More interestingly, even if some percentage of WRAP-PBNs was internalized by endocytosis-dependent mechanisms, they could be able to rapidly escape from endosomes, as suggested by leakage assays using large unilamellar vesicles (LUVs) reflecting the endosomal membrane composition.

Recently, we performed a structure activity relationship (SAR) study using the lead peptides WRAP1 and WRAP5 and 13 new analogues to gain more information about the relationship between the amino acid composition, nanoparticle formation, and cellular internalization of these siRNA-loaded peptides (manuscript submitted for publication).

The WRAP5 peptide was also shown to be a suitable gene delivery system in the context of cancer gene therapy, as shown by the WRAP5-mediated delivery of a p53 encoding plasmid (pDNA) [84]. Through the design of an experimental tool, the optimal ratio of nitrogen to phosphate groups (N/P) was determined for WRAP5:pDNA in comparison with the complex formed by the previously presented RALA peptide. In this context, both peptides were able to form PBNs in the presence of pDNA, with nearly identical zeta potential (~+33 mV) and pDNA complexation capacity (~90%), but with a smaller PBN size for WRAP5 compared to RALA (103.0 nm at N/P = 3 and 183.3 nm at N/P = 5, respectively).

## 7. C6 Family

CPPs complexing siRNA molecules to enable their cellular internalization have been, in some contexts, called “amino acid pairing” (AAP) peptides due to the two distinct domains responsible (i) for self-assembly and (ii) cell permeation [48]. Based on their amphipathicity, AAP peptide C6 (18mer) protected siRNA from RNAse through a non-covalent complexation. Prof. P. Chen’s group then developed the more water-soluble C6M1 peptide with a significantly reduced cell toxicity and an increased siRNA delivery in Chinese hamster ovary cells [49]. Furthermore, based on the higher amount of tryptophan residues within the sequence, C6M1 promoted endosomal escape once internalized via endosomal-dependent pathways. In the presence of 50% serum, C6M1 protected siRNA from serum RNase degradation over a period of 24 h, compared to 4 h for the naked siRNA [78]. Moreover, C6M1:siRNA reduced tumor growth through the silencing of the anti-apoptotic protein Bcl-2 after an intratumoral injection in mice [51].

To further promote the endosomal escape by the pH-buffering effect of protonable groups (pH sponge effect), Chen and colleagues introduced histidine residues into peptide of C6 and C6M1, creating seven new analogues (C6M2–C6M8) with histidine substitutions [52]. Furthermore, the peptides C6M6 to C6M8 were designed with an additional glycine residue at the N-terminal end. Such a modification has been reported to increase the stability and fusion activity of some CPPs [98]. Two peptides, C6M3 and C6M6, complexed with siRNA, achieved above 60% GAPDH gene expression silencing in CHO-K1 cell line. More importantly, they were able to reduce the anti-apoptotic Bcl-2 protein level, inhibiting tumor growth in a mouse xenograft tumor model after an intratumoral injection. Further investigation revealed that the more efficient stoichiometry to form complexes between C6M3 and siRNA was 7:1 (achievement of neutrality). However, better siRNA uptake was acheived with higher molar ratios (MR 20:1 and MR 40:1) due to stronger cell membrane interactions with the large excess of peptides [99].

## 8. Mgpe Family

Based on investigations modulating the amphipathicity and charges of several pVec analogues, Dr. M. Ganguli’s group modified the physicochemical parameters of the amphipathic peptide Mgpe-1, derived from human protein phosphatase 1E, to promote nucleic acid delivery [21,100]. The Mgpe family includes primary and secondary amphipathic peptides, mainly tested for plasmid delivery in different cell lines. Mgpe-3 and Mgpe-4 peptides displayed a high transfection efficiency, equivalent to that of commercial agents with a lower cytotoxicity and with stability in the presence of serum [21]. In addition, several developments have enabled the improvement of pDNA transfection efficacy. For example, the addition of cysteine increased the transfection efficiency of a secondary amphipathic Mgpe-9, and the coating of Mgpe/plasmid polyplexes with glycosaminoglycans such as chondroitin sulphate (CS) displayed the enhancement of polyplexes’ stability and pDNA delivery efficiency [23,54]. Recently, Ganguli and co-workers described that Mgpe polyplexes could also induce a high transgene expression in differentiated non-dividing cells, known to be difficult to transfect, and that an additional CS coating improved the diffusion of the polyplexes in the vitreous, suggesting the possibility of delivering genetic material to the retina [55].

## 9. Other CPPs Forming PBNs

In this chapter, a subset of CPPs used for oligonucleotide delivery by forming self-assembled nanoparticles was selected based on their potential therapeutic applications.

### 9.1. MPG

The 27mer primary amphipathic MPG peptide containing a hydrophobic domain (derived from the fusion sequence of HIV gp41) and a hydrophilic domain (derived from the nuclear localization sequence of SV40 T-antigen) was designed in the late 1990s for the delivery of oligonucleotides [20]. Since 2015, only a few papers were published using MPG as a unique oligonucleotide delivery system, and this peptide was used more as a grafted entity on PLGA polymers in order to increase their cellular translocation [101,102]. However, some recent works were recently published from Dr. A. Bolhassani’s group using MPG alone for the delivery of genes to develop an effective vaccine against the hepatitis C virus (HCV) [103] or human immunodeficiency virus (HIV) [104].

### 9.2. CHAT

In order to develop the ideal CPP for oligonucleotide delivery, Prof. H. McCarthy’s group designed a 15mer CHAT peptide forming PBNs (~200 nm) in the presence of pDNA [56]. This peptide was composed of arginine residues for nucleic acid complexation and cellular uptake, tryptophan to enhance hydrophobic cell membrane interactions, histidine to allow endosomal escape, and cysteine for stability to confer controlled intracellular cargo release through the reduction of disulphide bonds in the intracellular environment. Due to its impressive pDNA delivery in vitro (cell line) and in vivo (tissue), CHAT could be a new peptide for the delivery of nucleic acid therapeutics.

### 9.3. StA-TH

To design a gene delivery system entering cells in acidic solid tumors with minimal cellular uptake in normal tissues, Zhang and co-workers replaced first lysine residues of the TH peptide (an analog of TP10) by histidine moieties, and secondly attached a stearyl fatty acid chain at its N-terminus [57].

### 9.4. T9(dR)

This CPP is a 36mer peptide composed of transportan (TP—27mer) and a nona-D-arginine block (9(dR)—9mer) [58]. The chimeric T9(dR) peptide was designed for the knock-down of the nucleoprotein (NP) of the influenza virus as siRNA-based therapy. Despite its high length, T9(dR)-PBNs delivered siRNA into the respiratory tract (epithelial cells) of influenza-infected BALB/c mice, which induced the inhibition of influenza virus replication more efficiently than TP, thus alone revealing the importance of the additional poly-Arg sequence.

### 9.5. p5RHH

This peptide was derived from the cytolytic peptide, melittin, extracted from honey bee venom. By introducing some modifications, the cytotoxic properties of the peptide were reduced while maintaining its interactions with membranes and increasing those with siRNA for the complex formation [105]. More recently, the p5RHH peptide was used to efficiently and deeply transfect siRNA targeting NF-kB in human cartilage to prevent cartilage degeneration [59,60], as well as for siRNA-targeting TAM receptor tyrosine kinase family member AXL in xenografted ovarian and uterine cancer mice [61]. Furthermore, the p5RHH peptide was also used for miRNA transfection [62].

### 9.6. BR2

Issued from the buforin IIb antimicrobial peptide, the BR2 was designed to keep its cancer-specific toxicity and to reduce the cytotoxicity against normal cells [63]. First used in a covalent strategy for the delivery of a single-chain variable fragment (scFv) antibody against mutated K-ras, BR2 peptide was also able to form PBN in the presence of siRNA (~170 nm at N/P ratio of 8) [63]. Interestingly, BR2 peptide has the same RLLR motif within its sequence that is found in other CPPs able to form PBN, such as CADY [22], RICK [25], C6M1 [50], and WRAP [26].

### 9.7. S4(13)-PV

S4(13)-PV peptide was a chimera between the dermaseptin-derived peptide and the nuclear localization sequence of the SV40 large T antigen [106]. S4(13)-PV was successfully used to complex splice-switch oligonucleotides (SSOs), siRNA, or pDNA. However, these complexes were mainly entrapped in endosomes [107]. Significantly higher transfection efficiencies were obtained by associating cationic liposomes (lipoplexe formation). Escape from lysosomal degradation was achieved by adding a C12 lauryl chain to the S4(13)-PV peptide N-terminus, resulting in high lipid bilayer destabilization capacities, as well as efficient gene silencing [65]. The insertion of five histidine residues between the C12 chain and the S4(13)-PV peptide increased the homogeneity of the formed nanoparticles (better polydispersity index), resulting in an enhanced siRNA transfection for the downregulation of stearoyl-CoA-desaturase-1 overexpressed in cancer cells [65].

### 9.8. StA-SPA

The 11mer peptide Substance P (SP, RPKPQQFFGLM-NH2) with cell-penetrating properties [108] was modified to obtain the peptide called SPA ([DArg^1^, D-Trp^5,7,9^, Leu^11^]Substance P) [66]. SPA peptide was further optimized for efficient pDNA delivery by grafting a stearic acid to the N-terminus of the peptide, which enabled complex formation (>200 nm imaged by TEM), cellular internalization, and luciferase expression comparable to LF2000.

### 9.9. KL4

The 21mer peptide KL4 was designed based on the structural characteristics of surfactant protein B (SP-B) [109]. KL4 mediated siRNA transfection effectively through the formation of nanosized complexes in human lung epithelial cells (A549 and BEAS-2B cells) in a comparable way as performed by Lipofectamine 2000 [67]. More recently, Qiu et al. designed analogues of KL4 in order to obtain more soluble peptides (replacement of leucine by alanine or valine). However, these replacements impacted siRNA complexation due to the disruption of the α-helical structure of KL4, which, in turn, reduced siRNA transfection [110].

## 10. Functionalized PBNs

### 10.1. PEGylation

One major drawback of PBNs as an in vivo delivery system is their short life span in the blood circulation. Their size and their charge could influence the recognition by specific defense systems of the body, and then the absorption by the system of mononuclear phagocytes, which would prevent them from entering other tissues. To circumvent this limitation, PEGylation has been considered as a significant shielding strategy (Figure 2). Indeed, the PEGylation of nanoparticles has several pharmacological advantages such as improved drug solubility, increased drug stability, and an extended circulating life [111]. Moreover, reduced toxicity and rate of kidney clearance, enhanced protection from proteolytic degradation, decreased immunogenicity, and a minimal loss of biological activity might be also noticed when nanoparticles are PEGylated. Thus the grafting of polyethylene glycol (PEG) moieties improved their physical stability in vivo, while preventing both recognition by the mononuclear phagocytic system (MPS) in the liver and spleen and interactions with blood components [112]. Successful examples of PEGylated lipid-based nanoparticles are given by the FDA approved mRNA vaccines of BioNTech and Moderna [113] or siRNA therapeutic ONPATTRO^®^ for polyneuropathy in hereditary transthyretin-mediated (hATTR) amyloidosis [1].

In analogy to the lipidic PEGylation, CPPs were also PEGylated in order to enhance their in vivo application. Prof. Langel’s group used this approach to increase passive accumulation of their NF55 nanoparticles to tumors based on the PEG shielding effect, improving their half-life in serum and reducing renal clearance [64]. Due to a low extracellular pH and an important intracellular glutathione concentration for tumor cells, a pH- and glutathione-sensitive disulfide bond was introduced between the peptide and the PEG moiety to facilitate pDNA delivery. Indeed, the PEGylation of NF55 (= NF552) resulted in a reduced lung accumulation and threefold higher tumor accumulation using a formulation containing 20% PEGylated peptide. In a similar way, our group reported that a low PEGylation of RICK (20%) did not alter nanoparticle formation, cellular internalization, or the silencing efficiency of PBNs [46]. Moreover, we could clearly demonstrate that 20% PEGylated RICK-PBNs revealed a higher biodistribution in zebrafish embryos injected at the one-cell stage, as well as reduced liver and kidney accumulation in mice after intravenous injection. As 100% PEGylation has a negative impact on the efficiency of cellular siRNA delivery with PBNs, most of developments involved a low PEGylation ratio [44,46].

Another example was given with the multi-domain FLR peptide composed of an HS-binding sequence derived from fibroblast growth factor 2 (FGF2), the pan-nucleic acid interaction sequence LK15, and the poly-Arg CPP 8R designed for nucleic acid delivery [114]. With PEGylation rates ≥40%, the positive surface charge of the nanoparticles (100 nm–140 nm) maintained their hydrodynamic size in bronchoalveolar lavage fluid (BALF). More importantly, PEGylated particles showed superior biodistribution and efficient pDNA transfer compared to non-PEGylated complexes in healthy mouse lung models.

After being conjugated with PEG, the gene delivery systems showed reduced in vivo specificity based on steric hindrance, therefore Prof. Chen’s group grafted a short diethylene glycol (DEG), instead of PEG, to the C6M1 peptide (= DM1 peptide) [53]. DM1:siRNA complexes showed remarkable serum stability without changing the gene silencing properties as measured by mRNA and protein quantifications.

PEGylation could be also used to facilitate the solubilization of a hydrophobic KL4 peptide by attaching a monodisperse linear PEG of 12mers [115]. The PEG_12_-KL4 peptide formed nanosized complexes with mRNA at a 10:1 ratio (*w*/*w*), and mediated effective transfection on lung epithelial cells in vitro and in vivo after an intratracheal administration to mice.

### 10.2. Cell/Organ Targeting

Cell and/or organ targeting was achieved by grafting short peptides known to recognize overexpressed receptors on the cell surface onto nanoparticles (Figure 2) [116]. In some cases, it was also possible to target a specific organ, as reported first for the NF55:pDNA nanoparticles [43]. Based on this result, Kurrikoff and colleagues performed in-depth analysis using PF14- and NF55-PBNs for the specific lung delivery of siRNA and pDNA using mice with acute lung inflammation and asthma [89]. Important anti-inflammatory effects were recorded in both disease models using siRNA targeting cytokine TNFα, resulting in decreased disease symptoms. This finding was surprising because Freiman et al. showed in a previous publication that the PEGylation of NF55 (= NF552) revealed a reduced lung accumulation [44].

Prof. S.N. Bhatia’s group has been working for many years on the development of tumor-penetrating nanocomplexes (TPN) composed of a CPP (Transportan), a fatty acid (Myristoyl), and a tumor targeting peptide (Tp-Lyp-1) [117,118]. These siRNA loaded TPNs entered in the cytosol via a receptor-specific fashion and could be used to target ovarian cancer. Furthermore, the slightly modified TPNs, by changing the Lyp-1 targeting sequence with iRGD, were demonstrated to deliver siRNA to pancreatic cancer [119], as well as single chimeric guide (sgRNA)/Cas9 protein complex inside cells [120].

### 10.3. Organelle Targeting

Dysfunctions at the organelle level are known to be implicated in several diseases (e.g., lysosomal storage disease or peroxisomal disorder), making organelle targeting essential. In this context, the easy chemical modification of CPPs in many different ways should ensure the specific therapeutic delivery into these intracellular organelles such as mitochondria, lysosomes, or the nucleus (Figure 2) [121].

For nucleus-targeting, CPPs were coupled to nuclear localization sequences (NLSs) [122]. These lysine-, arginine-, or proline-rich motifs recognize importin, a type of karyopherin that transports proteins from the cytoplasm to the nucleus, thus facilitating the nuclear import and localization of the gene carriers. Nowadays, NLS sequences are mainly grafted onto polymeric nanoparticles (e.g., PLGA), as described by Yameen et al. [123].

For lysosomal targeting, it was possible to graft lysosomal sorting peptides (LSP), often composed of short tyrosine-based peptide sequences of 4–5 amino acids, onto the CPP sequence [124]. Specific lysosomal delivery was shown using Tat-derived gold nanoparticles [125], but, to our knowledge, this strategy has never been applied for oligonucleotide PBN delivery.

For mitochondrial-targeting, several strategies have been developed. For example, Kelley and colleagues presented mitochondria-penetrating peptides (MPPs) [126,127], which were mainly used for the covalent delivery of different cargoes [128]. A bit earlier, Szeto and Schiller introduced cell-permeable, mitochondrial-targeted peptides [129,130], which were grafted onto different polymer-based nanoparticles, but not onto PBNs. Based on the Szeto–Schiller peptide SS-31, Prof. Langel’s group developed a set of mitochondrial-penetrating peptides based on the covalent fusion of PF14 and mtCPP1 [131] for antisense oligonucleotide (ASO) delivery [132]. In this report, the specific delivery of peptide/oligonucleotide nano-complexes were shown as proof-of-principle for the potent therapeutic application to patients with mitochondrial diseases.

Peptide-based nanoparticles (PBNs) could be functionalized by PEGylation, acylation, or by grafting cell or organelle targeting sequences. For each PBN modification, some advantages or disadvantages are provided.

For Golgi and endoplasmic reticulum (ER) apparatus targeting, oligonucleotide-loaded PBNs could be grafted with the ER retention four-peptide sequence KDEL, as described by Jian Zhang et al. for the apoptosis-inducing fusion peptide TAT-IL-24-KDEL [133]. However, to our knowledge, no PBN grafting has been reported up to now.

## 11. Conclusions

Since the first identification of cell-penetrating peptides (CPPs) as potential new delivery systems, a lot of work has focused on their use in nucleic acid (NA) delivery through a non-covalent strategy consisting of the formation of stable peptide-based nanoparticles (PBNs). Although it is quite difficult to classify PBNs by the function of their physico-chemical properties, their different NA cargoes, or their in vitro/in vivo applications, some main common rules have been identified.

Probably the first contacts during PBN formation result from the electrostatic interactions between the positive charges of the CPP (arginine/lysine/histidine) and the negative charges of the NAs (phosphate groups). However, the presence of cationic residues within the peptide sequence is not sufficient to support self-assembly in the presence of the cargo; hydrophobic domains are also required for CPP:NA complex formation. These hydrophobic contributions might come from hydrophobic amino acids, as well as from the insertion of fatty acid chains in most of the cases at the N-terminus of the CPPs. This dual nature of PBN-forming peptides is often associated with an amphipathic feature resulting from the primary or secondary structure, but can also be extended to specific 3D structures favoring the condensation of NAs with positive charges and self-assembly in nanoparticles. As the peptides’ positive charges are crucial for NA complexation, they also condition the final surface charge of the PBNs. As described here, most of the developed PBNs displayed a final positive zeta potential in the range of +10 mV to +40 mV, depending on the used conditions (ratio, solvent). Although there is still a debate in the field of nanomedicine with regard to the surface charge of therapeutic nanoparticles and its consequence on their opsonization in the bloodstream, most of the PBNs were mainly described as positively charged, able to deliver NAs in the presence of serum, suggesting that the positive surface charge is not a limitation in vitro. However, in order to determine the correct zeta potential, some experiment conditions should be respected, because the presence of salt ions in the vicinity of charged particles will manifest itself in two ways: (i) the same particle prepared in a buffer with less salt will have a higher (absolute) zeta potential, and (ii) the same (molar) concentration of a higher valency salt will have a stronger effect on zeta potential [134]. Therefore, we recommend always performing the zeta potential measurement in a solution containing a low concentration of monovalent ions (e.g., 5 mM NaCl), and not in pure water.

Moreover, with regard to an in vivo application, the stability of PBNs in the presence of physiological conditions (serum, blood, etc.) was shown to be crucial. Indeed, the stability of PBNs directly impacts their size and homogeneity, and, like any nanomedicine, it is essential that the PBNs were homogeneous, with a size in the range of 60 nm to 150 nm and with a polydispersity index (PdI) less than 0.3 (monodispersed distribution) (Table 1). The size of PBNs directly determines their surface area interacting with biological environments, thus influencing their blood circulation time and their biodistribution. PBNs smaller than 6 nm were filtered out by the kidneys, whereas PBNs larger than 200 nm can be rapidly captured by the liver and spleen due to the activation of complement [135]. The homogeneity of PBNs with reproducible PdI measurements is also required for a clinical application, and proposed complexes with a size of 152 nm but with a PdI of 0.68 [135] will probably not be used for further therapeutic development. Therefore, to achieve the optimal conditions for the design of PBNs, we recommend the imperative measurement of the PdI during the first biophysical characterization to assess homogeneity, which is an important requirement for clinical applications.

PBNs are well-suited NA delivery systems, and could be used as versatile tools in biomedicine. Compared to other carriers, CPPs have a low cell cytotoxicity and could be easily degraded into amino acids, and are therefore suitable for preclinical and clinical studies. However, despite their unprecedented efficiency in delivering therapeutic cargos into cells, CPP-mediated strategies are still not used in clinical applications. In order to push forward the clinical translation, several features are to be taken into account. First, the stability, size, and monodispersity of the PBNs should be controlled, and could be optimized by grafting PEG motifs or fatty acids to the CPPs, as reported by different publications [44,46,115]. Indeed, the PEGylation of nanoparticles reduces their size and then their opsonization, minimizing their clearance by the reticuloendothelial system (RES) and leading to longer blood circulation times and improved pharmacokinetic properties [136]. The use of grafted shielding groups, such as polysaccharides and PEGs, also reduce surface charges, leading to neutrally charged particles, which thus have a much lower opsonization rate than charged particles [137]. Secondly, the specificity of PBN-based internalization could be improved by grafting targeting (or homing) sequences recognizing specific receptor overexpressed in cancer cells [138] or on cellular organelles [121]. Unlike passive targeting consisting of the accumulation of the nanoparticles in the liver and lungs or in tumors through their enhanced permeability and retention effects, active targeting requires the appropriate ligand molecules in order to drive the nanoparticles to the specific organ or tumor site. Thus, finding the specific ligand to graft to PBN is also a key point for their clinical application.

Finally, an impressive work has also been performed on the development of stimulus-responsive “smart” CPP-based systems, which could be pH- or enzyme- triggered [139]. More specifically, as well as naked PBNs, the engineered “smart” PBNs should penetrate through many physiological barriers without inducing undesirable host immune responses or losing its colloidal stability after intravenous injection and reaching the most effective delivery into targeted cells.

## Figures and Tables

**Figure 1 biomedicines-09-00583-f001:**
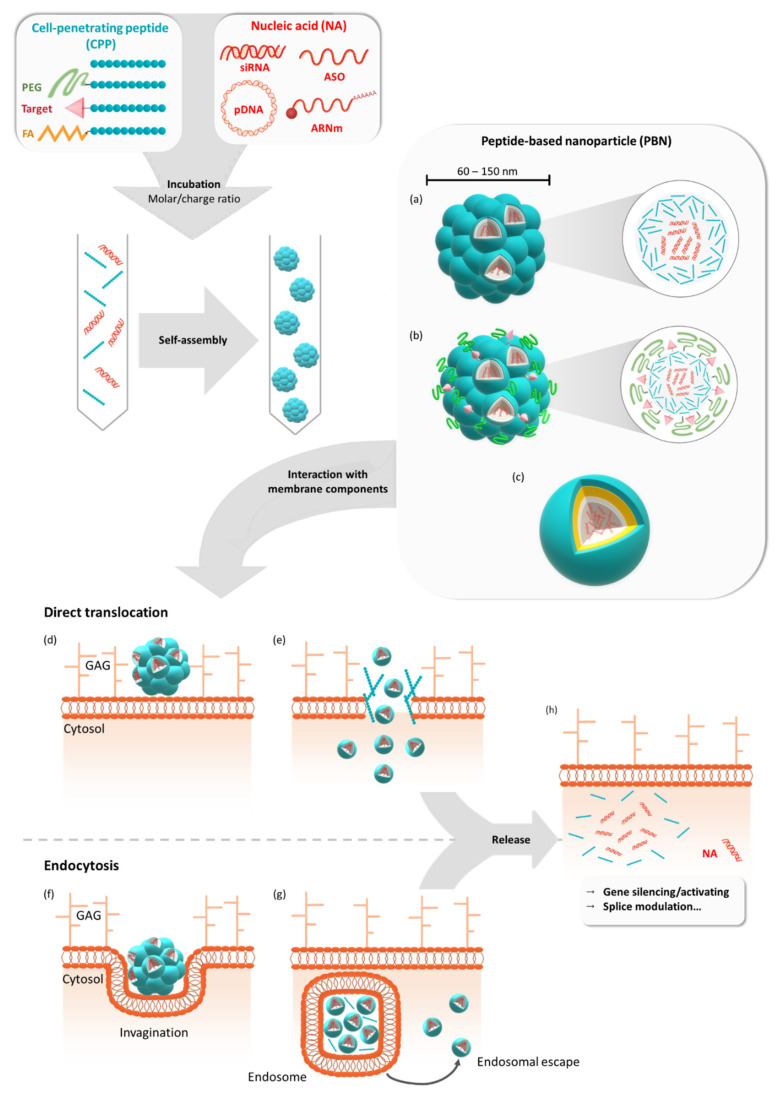
Formulation of peptide-based nanoparticles in the presence of different nucleic acids and their cellular internalization. Peptide-based nanoparticles (PBNs) are formulated by mixing a cell-penetrating peptide (CPP) or a grafted CPP (PEGylated, targeting sequence or fatty acid) with a nucleic acid (NA) such as pDNA, mRNA, siRNA, or ASO at a given molar or charge ratio. By mixing these two compounds, the nanoparticle is formed by self-assembling into naked PBNs (**a**), a multi-grafted PBNs (**b**), or prospective micelle-like PBNs (no model available) (**c**). In all cases, the PBNs of mean size between 60 nm and 150 nm encapsulate several NAs for cellular transfection. Thereafter, cellular internalization could occur via direct translocation (**d**) or via endocytosis-dependent pathways (**f**). After the direct translocation (**e**) or endosomal escape (**g**), the NAs could be active either by silencing or activating genes or by performing splice modulation (**h**). GAG = glycosaminoglycans.

**Figure 2 biomedicines-09-00583-f002:**
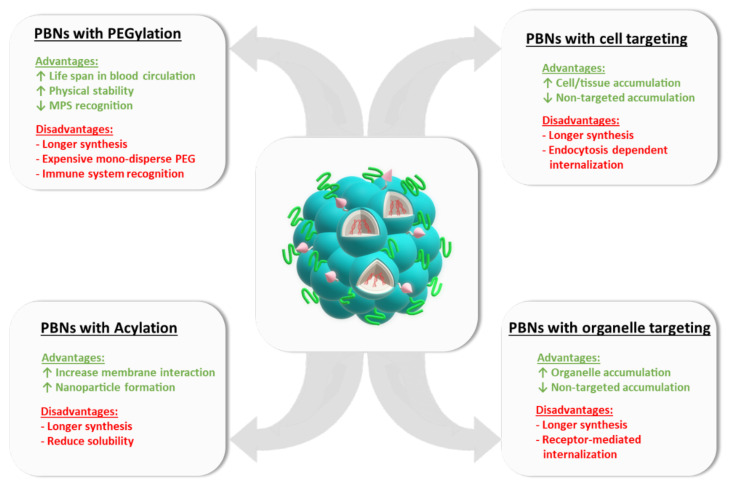
Examples of functionalized PBNs.

**Table 1 biomedicines-09-00583-t001:** Cell-penetrating peptides used for nucleic acid transfection through peptide-based nanoparticle (PBN) formation.

Name	Sequence	Cargo	Ratio	Size (nm)	PdI	ZF (mV)	In Vitro Activity		In Vivo Activity	Ref
							Cells	Effect		
Poly-Cationic Familly										
**StA-R8**	Stearyl-RRRRRRRR	siRNA	CR—4:1	185.2	n.d.	15.6	HepG2, A549	60% survivin KD	n.d.	[31]
**OA-R8**	Oleyl-RRRRRRRR			191.9		13.2				
**K9[Ca^2+^]**	KKKKKKKKK	pDNA	N/P—10	200–400	n.d.	~ +20 (1 mM KCl)	HeLa, A549, HEK-293, LLC, MDA-MB-231	Higher pLuc expression as PEI	Mice lung tumor	[32]
**R9[Ca2+]**	RRRRRRRRR	pDNA	N/P—10	200	n.d.	~ +20 (1 mM KCl)	A549, HEK-293	Higher or equal pLuc expression as PEI	n.d.	[33]
**RH9[Ca2+]**	RRHHRRHRR					~ +12 (1 mM KCl)				
**RA9[Ca2+]**	RRAARRARR					~ +8 (1 mM KCl)				
**RL9[Ca2+]**	RRLLRRLRR									
**RW9[Ca2+]**	RRWWRRWRR					~ +10 (1 mM KCl)				
**GALA/KALA/RALA familly**										
**RALA**	WEARLARALARALARHLARALARALRACEA-C	pDNA	N/P—10	51	0.35	+29 #	ZR-75-1, PC-3, NCTC-929	eGFP expression not better than Lipofectamin but less toxic	pLuc expression in the lungs and liver of mice	[34]
		siRNA	N/P—10	∼55–65	<0,60	∼ +20–25 #	ZR-75-1	Equal FKBPL KD compared to Oligofectamin but less toxic	RALA:siFKBPL has no effect on tumor growth	[35]
		siRNA	N/P—6	76.6	n.d.	+16.5 #	HMEC-1	Efficient FKBPL KD	RALA:siFKBPL in wound patches increases wound healing in mice	[36]
		siRNA	N/P—9	100–110	<0.35	~ +38 #	hDF (2D and 3D), THP-1 derived macrophages	Efficient MMP-9 KD	n.d.	[37]
		mRNA	N/P—10	91	n.d.	+26.3 #	DC2.4	Expression of eGFP	Increased T cell response compared to DOTAP transfection	[38]
**RGSG**	WEGRSGRGSGRGSGRHSGRGSGRGSRG-C	mRNA	N/P—10	150	n.d.	+2 #	DC2.4	No eGFP expression compared to RALA	Less T cell response compared to RALA	[38]
**RRRR**	WEGRRRRRRR-C	mRNA	N/P—10	1050		−5 #				
**PepFect/NickFect familly**										
**PF14**	Stearyl- AGYLLGKLLOOLAAAALOOLL	SCO	MR—5:1	363	n.d.	−28.4 (0.01 mM KCl)	HeLa pLuc705, U2OS, mdx mouse myotubes	Equal or better splice correction compared to Lipofectamin	n.d.	[39]
		mRNA	N/P—3	92	0.248–0.259	n.d.	SKOV-3 (2D and 3D)	eGFP expression Lower in 2D but higher in 3D compared to Lipofectamine MessengerMax	mCherry expression in xenografted mice	[40]
		SCO	MR—5:1	295.3 *	0.732 *	n.d.	HeLa pLuc 705	Internalization in comparison with small molecules	n.d.	[41]
**PF14**	Stearyl- AGYLLGKLLOOLAAAALOOLL	pDNA	N/P—2	~150	n.d.	~ +35 #	CHO	Dose-dependent pLuc expression	pLuc expression in the lungs and liver of mice	[42]
**PF14-O**	Stearyl-AGYLLGKLLOOLAOOALOOLL	pDNA	N/P—2	125	n.d.	~ +32 #	CHO	pLuc expression with PF14-O better than PF14-E	pLuc expression in the lungs and liver of mice	
**PF14-E**	Stearyl-AGYLLGKLLEOLAAAALOOLL			125	n.d.	~ +35 #			n.d.	
**C0-PF14**	AGYLLGKLLOOLAAAALOOLL			1500	n.d.	~ +8 #		Nearly no pLuc expression	n.d.	
**C10-PF14**	Decanoyl-AGYLLGKLLOOLAAAALOOLL			100	n.d.	~ + 22 #		pLuc transfection with C10-PF14 lower than C22-PF14	n.d.	
**C22-PF14**	Docosanoyl-AGYLLGKLLOOLAAAALOOLL			125	n.d.	~ +40 #			pLuc expression in the lungs and liver of mice	
**C22-PF14-O**	Docosanoyl-AGYLLGKLLOOLAOOALOOLL			100	n.d.	~ +30 #		pLuc transfection of C22-PF14 and C22-PF14-O equivalent to PF14 and better that C10-PF14	pLuc expression in the lungs and liver of mice	
**NF53**	(Stearyl-AGYLLG)ε-KINLKALAALAKKIL	pDNA	CR—3:1	74.3	0.360	−14.9 (OptiMEM + 10% FBS)	CHO	Equal eGFP expression compared to LF200	n.d.	[43]
		SCO	MR—10:1	135.3	0.459	−8.8 (OptiMEM + 10% FBS)	HeLa pLuc 705	Lower splice correction compared to LF200		
		siRNA	MR—10:1	68.6	0.529	−11.9 (OptiMEM + 10% FBS)	EGFP-CHO	Higher eGFP silencing compared to RNAiMax		
**NF61**	Stearyl-AGYLLGKINLKALAALAKKIL	pDNA	CR—3:1	68.7	0.200	−17.9 (OptiMEM + 10% FBS)	CHO	Equal eGFP expression compared to LF200		
		SCO	MR—10:1	60.5	0.286	−10.2 (OptiMEM + 10% FBS)	HeLa pLuc 705	Lower splice correction compared to LF200		
		siRNA	MR—10:1	159.4	0.348	−13.6 (OptiMEM + 10% FBS)	EGFP-CHO	Equal eGFP silencing compared to RNAiMax		
**NF51**	(Stearyl-AGYLLG)δ-OINLKALAALAKKIL	pDNA	CR—3:1	62	0.138	−11.5 (OptiMEM + 10% FBS)	CHO, MEF, Jurkat, A20	Higher eGFP expression compared to LF200		
		SCO	MR—10:1	86.0	0.298	−11.1 (OptiMEM + 10% FBS)	HeLa pLuc 705	Higher splice correction compared to LF200		
		siRNA	MR—10:1	74.2	0.197	−11.8 (OptiMEM + 10% FBS)	EGFP-CHO	Higher eGFP silencing compared to RNAiMax		
**NF51**	(Stearyl-AGYLLG)δ-OINLKALAALAKKIL	pDNA	CR—4:1	n.d.	n.d.	n.d.	HeLa, U87-MG, N2A and HT1080	eGFP expression comparable to LF200, NF55 better than NF51 and NF54	pLuc expression in the lung, liver, and brain of healthy mice and those bearing intracranial tumors	[44]
**NF54**	(Stearyl-AGYLLG)δ-OINLKALAALAAKIL			n.d.						
**NF55**	(Stearyl-AGYLLG)δ-OINLKALAALAKAIL			50–150						
**NF55**	(Stearyl-AGYLLG)δ-OINLKALAALAKAIL	pDNA	CR—4:1	85 #	0.211	n.d.	CHO	pLuc expression comparable to Freiman 2016	pLuc lung expression	[45]
**CADY-K/RICK family**										
**CADY-K**	GLWRALWRLLRSLWRLLWK	siRNA	MR—20:1	116	0.30	+38.0 (5 mM NaCl)	U87, Neuro2A, B16	Efficient Luciferase and CyclinB1 proteins KD	n.d.	[25,28]
**d-Cady-k**	glwralwrllrslwrllwk	siRNA	MR—20:1	90	0.30	+40.0 (5 mM NaCl)	U87 #			
**RICK**	kwllrwlsrllrwlarwlg	siRNA	MR—20:1	92	0.24	+40.0 (5 mM NaCl)				
**PEG-RICK**	PEG_2000_-Ckwllrwlsrllrwlarwlg	siRNA	MR—20:1	69 (20% PEG)	0.29 (20% PEG)	+37.0 (20% PEG) (5 mM NaCl)	U87	Efficient Luciferase and CDK4 proteins KD for 20% PEG-RICK NPs and less cytotixicity than RNAimax.	20% PEG-RICK NPs significantly reduce liver and kidney accumulation in mice	[46]
**WRAP family**										
**WRAP1**	LLWRLWRLLWRLWRLL	siRNA	MR—20:1	73.3	0.38	+42.2 (5 mM NaCl)	U87, KB, MCF7, HuH7, Neuro2A, MDA-MB-231, CMT93, HT29, RM1, GL261	Efficient Luciferase or CDK4 KD, fast internalization and less toxic than RNAimax	n.d.	[26]
**WRAP5**	LLRLLRWWWRLLRLL	siRNA	MR—20:1	80.0	0.29	+28.8 (5 mM NaCl)				
		pDNA	N/P—3	102	n.d.	+33 (Tris buffer pH 7)	n.d.	n.d.	n.d.	[47]
**C6 family**										
**C6**	RLLRLLLRLWRRLLRLLR	siRNA	MR—40:1	150–250	n.d.	+60.0 #	CHO-K1	Internalization of fluorescently labelled siRNA without cytotoxicity	n.d.	[48]
**C6M1**	RLWRLLWRLWRRLWRLLR	siRNA	MR—30:1	~70	n.d.	+31 (HEPES) +5 (PBS)	CHO-K1	Better internalization of fluorescently labelled siRNA than C6, significant inhibition of GAPDH expression	n.d.	[49,50]
		siRNA	MR—60:1	~100–200	n.d.	~ +50 #	CHO-K1	Significant inhibition of GAPDH expression	Inhibition of tumor growth with Bcl-2 siRNA in A549 cancer cells xenografted in mice	[51]
**C6M3**	RLWHLLWRLWRRLHRLLR	siRNA	MR—40:1	~90	n.d.	+32.0 #	CHO-K1, RAW 264.7	Strong uptake CHO-K1 cells, significant inhibition of GAPDH expression and no significant cytokine induction	Inhibition of tumor growth with Bcl-2 siRNA in A549 cancer cells xenografted in mice	[52]
**C6M6**	GLWHLLLHLWRRLLRLLR	siRNA	MR—60:1	~110	n.d.	+36.0 #				
**DM1**	DEG-RLWRLLWRLWRRLWRLLR	siRNA	MR—40:1	n.d.	n.d.	n.d.	CHO-K1, C166-GFP	Significant inhibition of GAPDH and eGFP expression and DEGylation improves serum resistance	n.d.	[53]
**Mpge family**										
**Mgpe-1**	SRLSHLRHHYSKKWHRFR	pDNA	N/P—10	80.13	0.151	+36.5 #	CHO-K1, MCF-7, A549	Equal pLuc expression as Lipofectamine but less toxic and higher pLuc expression than Cellfectin/Superfect	n.d.	[21]
**Mgpe-2**	LLYWFSRSHRHHSKKHRR		N/P—10	110.25	0.141	+31.95 #				
**Mgpe-3**	RRLRHLRHHYRRRWHRFR		N/P—10	63.26	0.152	+33.5 #				
**Mgpe-4**	LLYWFRRRHRHHRRRHRR		N/P—5	62.84	0.155	+25.45 #				
**Mgpe-10**	CLLYWFRRRHRHHRRRHRRC	pDNA	N/P—10	128.49	0.166	+27.7 #	CHO-K1, MCF-7, A549, B16, B35, H 1299, HEK, Jurkat, MDA-MB-231, RAW, U87, T47D, Hela	Higher transfection efficiency, less toxic than Lipofectamine and Chondroitin sulfate combination	n.d.	[23,54]
**Mgpe-9**	CRRLRHLRHHYRRRWHRFRC	pDNA	N/P—10	85.77	0.240	+35.5 #				
		pDNA	N/P—10	50.63	n.d.	+24.0 #	dividing/differentiated ARPE-19/hfRPE cells	eGFP and Gaussia Luciferase expression	n.d.	[55]
		siRNA	N/P—10	173.9	n.d.	n.d.	Differentiated ARPE-19	80% GAPDH knockdown GAPDH for polyplexes at N/P 30 and combined with condroitin sulphate	n.d.	
**Other PBN-forming peptides**										
**CHAT**	CHHHRRRWRRRHHHC	pDNA	N/P—12	207	0.25	+29 #	MCF7, MDA-MB-231, DU-145, PC-3	eGFP expression comparable to RALA	10-fold increased pLuc expression in lung, liver and kidneys, 5-fold in tumor	[56]
**StA-TH**	Stearyl- AGYLLGHINLHHLAHL (Aib)HHIL	pDNA	N/P—3	>200 (TEM)	n.d.	n.d.	CHO, U251	High internalization and p53 activity (pro-apoptotic) at pH 5.5	n.d.	[57]
**T9(dR)**	GWTLNSAGYLLGKINLKALAALAKKIL-(dR)9	siRNA	MR 4:1	350–550	n.d.	n.d.	293T, MDCK, RAW, A549	Silencing of nucleoprotein expression	Better survival and weight recovery of PR8 influenza viru-infected mice	[58]
**p5RHH**	VLTTGLPALISWIRRRHRRHC	siRNA	MR—100:1	~55 (TEM)	n.d.	n.d.	/	/	Silencing NF-kB expression reduced chondrocyte apoptosi in a murine model of controlled knee joint impact injury	[59]
						~ +12 #	HUVEC	p65 slencing	n.d.	[60]
				n.d.	n.d.	n.d.	ARK1, OVCAR8	AXL silencing	Reduced tumor nodules and weight	[61]
		mRNA	350 ng mRNA:2.0 nmol p5RHH	<200	n.d.	+6 (OptiMEM)	B16F10, CASMC, HAoEC	RFP, Luc, GFP expression	RFP expression on injured femoral artery. mRNA construct (p27-miRNA-126-3p) prevents restenosis in a femoral artery wire injury mouse model	[62]
**BR2**	RAGLPFQVGRLLRRLLR	siRNA	N/P—8	150–200	n.d.	~ +10 #	HeLa, HCT116, HaCat, NIH3T3	GFP nd VEGF silencing comparable to PEI	n.d.	[63]
**C18-S4_13_-PV**	Stearoyl-ALWKTLLKKVLKAPKKKRKVC	siRNA	CR—2	~250	n.d.	~+12 (HBS-2)	U87	No significant GFP silencing	n.d.	[64]
**C16-S4_13_-PV**	Palmitoyl-ALWKTLLKKVLKAPKKKRKVC			~250		~+10 (HBS-2)		No significant GFP silencing		
**C14-S4_13_-PV**	Myristoyl-ALWKTLLKKVLKAPKKKRKVC			350		~ +10 (HBS-2)		GFP silencing		
**C12-S4_13_-PV**	Lauroyl-ALWKTLLKKVLKAPKKKRKVC			750		+8 (HBS-2)		GFP silencing		
		siRNA	CR—5	192	0.44	+23.6 (HBS)	U87, HeLa	GFP silencing lower than LF2000	n.d.	[65]
**C12-H_5_-S4_13_-PV**	Lauroyl-HHHHH-ALWKTLLKKVLKAPKKKRKVC			173	0.24	+22.6 (HBS)		GFP silencing equal to FL2000		
**H_5_-S4_13_-PV-C12**	HHHHH-ALWKTLLKKVLKAPKKKRKVC-Lauroyl			184	0.69	+19.7 (HBS)				
**StA-SPA**	Stearyl-rPKPwQwFwLL	pDNA	N/P—2	>200 (TEM)	n.d.	n.d.	CHO	Nearly equal pLuc expression as LF2000	n.d.	[66]
**KL4**	KLLLLKLLLLKLLLLKLLLLK	siRNA	(*w*/*w*)—20:1	280	0.28	n.d.	A549, BEAS-2B	Reduced GAPDH expression comparable to Lipofectamin 2000	n.d.	[67]

Footnotes: * values measured in serum-containing medium, # zeta potential measured in H_2_O, n.d. = not determined.

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
