# Peer review of "Peptide-Based Nanoparticles for Therapeutic Nucleic Acid Delivery"

_biomedicines, 2021, doi:10.3390/biomedicines9050583_

Round 1
Reviewer 1 Report
The review Peptide-based nanoparticles for therapeutic nucleic acid delivery presents an extensive literature research regarding conjugation of cell-penetrating peptides (CPPs) with nucleic acids (NAs) for the purpose of drug delivery. This review showed the main works on this field from 2015-2021, presenting different CPP families used for PBN formulation as well as functionalization and bioconjugation techniques. Overall, the review has a high scientific thoroughness and is well structured. This review differs from others that address the same thematic as https://doi.org/10.1016/j.addr.2016.06.008 and doi: 10.3390/molecules25153482, due to the breadth of CPP studied and the bioconjugation strategies as a way to improve current therapies, which is a great positive aspect. As main criticisms I highlight the English mistakes and lack of care in writing and mistakes concerning the references. I also consider that the review is a bit long and lacks images illustrating the covered topics. Furthermore, a comparison should be made between the different functionalization strategies in the format of a table or swot analysis.
Revisions must be made to accept the article for publication in Biomedicines.
Major critics:
(1) An English revision must be performed on the whole document for approval in Biomedicines. Some example of errors: "fan of different biomolecules", Table 1 - "silening" "an murine", "StA-R8:pDNA particle were bigger", "agent for delivery", "funcionalization was clearly topple over the direct", "disulphide bonds was introduced", among others.
(2) There are two major problems with the references: (i) there are a high number of references <2015, which are out of the review's scope, (ii) some references are not suitable, e.g. ref 1 is not the most indicated, since is about potentially disease-modifying Huntington drugs in development and not about RNA interference therapies for hATTR.
Minor corrections:
(1) Abstract: The abstract must be rewritten to better explain the highlights of the review. Why are NA-based therapies more universal? What you intend to say with that statment? Choose a more appropriate word rather than "powerful" for delivery systems. Why those formulation conditions are important? That is not clear for the reader.
(2) Keywords: cellular trafficking is not a representative word of the review, please change for another one.
(3) Explain why peptide-based vectors are inexpensive for the delivery of therapeutic NAs? (lines 90-91, page 2).
(4) Please change PEG moieties to PEGylated in Figure 1 caption;
(5) Please rewrite the first sentence of Poly-cationic family topic (lines 123-124) and better introduce that topic.
(6) PEGylation section: There are other advantages regarding PEGylation of PBNs, please explain them in the beggining of the topic. Three important works regarding PEGylation of CPPs should be cited in the review: doi: 10.2174/1568026620666200128142603 (general improvements through PEGylation for CPPs), dx.doi.org/10.1590/s2175-97902018000001009 (general improvements through PEGylation for biobetters) and doi.org/10.1039/C9TB00590K (thermostability enhancement)
Author Response
Please also see the attachment
Review Report Form
Open Review REVIEWER 1
Comments and Suggestions for Authors
The review Peptide-based nanoparticles for therapeutic nucleic acid delivery presents an extensive literature research regarding conjugation of cell-penetrating peptides (CPPs) with nucleic acids (NAs) for the purpose of drug delivery. This review showed the main works on this field from 2015-2021, presenting different CPP families used for PBN formulation as well as functionalization and bioconjugation techniques. Overall, the review has a high scientific thoroughness and is well structured. This review differs from others that address the same thematic as https://doi.org/10.1016/j.addr.2016.06.008 and doi: 10.3390/molecules25153482, due to the breadth of CPP studied and the bioconjugation strategies as a way to improve current therapies, which is a great positive aspect. As main criticisms I highlight the English mistakes and lack of care in writing and mistakes concerning the references.
Among the huge number of delivery systems based on cell-penetrating peptides (CPPs), we decided to focus our review on nanoparticles formed or generated only by the self-assembly of the peptide itself in the presence of nucleic acids. The peptides reviewed here are essentially cell-penetrating peptides but could be also other peptide families (Fusogenic, modified, grafted...).
We have deliberately avoided to include CPPs used in the covalent delivery strategy or CPPs used to decorate other nanoparticles such as polymers, nanotubes or even virus.
In order to clarify the focus of the review, we have changed the end of the introduction:
Page 4 lines 121-133 : “In this Review, we present different peptide families used for PBN formulation in the presence of various NAs (Table 1) which were used and published during the period between 2015 and 2021 (even if their design was reported years previously). Especially, we have focused this review on peptide-based nanoparticles formed by self-assembly of peptides which are essentially native or modified CPPs (PEGylated, grafted to fatty acids or fusogenic moieties, etc.). CPPs used in the covalent strategy or to decorate other nanoparticles such as polymers, nanotubes or even virus are reported elsewhere [29,30].
In details, we have first summarized seven “main” CPP families such as the poly-cationic, GALA/KALA/RALA, PepFect/NickFect, CADY-K/RICK, WRAP, C6, Mpge families and secondly highlighted new developed CPPs with high potential (see section “Other CPPs forming PBNs”). Finally, we have recapitulated known PBN optimization such as PEGylation and different targeting strategies (Figure 2) which are important for the development of “intelligent” PBNs in view of a pharmacological application.”
I also consider that the review is a bit long and lacks images illustrating the covered topics. Furthermore, a comparison should be made between the different functionalization strategies in the format of a table or swot analysis.
We have added a second figure (Figure 2, see attachment) illustrating the advantages and disadvantages of the PBNs modifications (PEGylation, Acylation, Targeting).
However, we decided to illustrate this review only with schemes because:
- Based on the multitude of PBNs applications, it is not easy to privilege one over the others.
- Figures showing published results from other authors needs publication agreements with the corresponding fees.
Major critics:
(1) An English revision must be performed on the whole document for approval in Biomedicines. Some example of errors: "fan of different biomolecules", Table 1 - "silening" "an murine", "StA-R8:pDNA particle were bigger", "agent for delivery", "funcionalization was clearly topple over the direct", "disulphide bonds was introduced", among others.
We have considered comments of Reviewer 1 by improving the English language through the whole manuscript. All changes were noticed in red in the manuscript.
(2) There are two major problems with the references: (i) there are a high number of references <2015, which are out of the review's scope, (ii) some references are not suitable, e.g. ref 1 is not the most indicated, since is about potentially disease-modifying Huntington drugs in development and not about RNA interference therapies for hATTR.
We thank the reviewer 1 for this comment. We are conscious that some of references are older than 2015. However we did not want to forget the original works that led to recent publications of these last five years. We are convinced that, even older than 2015, these references are in the scope of the review since inseparable of the more recent developments. For example, we could not cite the recent development of RALA for mRNA vaccination (Cole et al. 2019) without citing the original design of GALA (Li et al. 2004) and RALA (McCarthy et al. 2014). Moreover, with regard to the field of peptide-based nanoparticles formed by peptides self-assembly, we found impossible to exclude some contributors (Ganguli or Chen’s group) or specific peptides (NF51) only for a gap of 2 or 3 years older than 2015 (see Table 1). We did this subjective choice thinking to the data provided to the readers and making an effort to forget none.
We corrected the reference 1 by Hoy, S.M. Patisiran: First Global Approval. Drugs. 2018, 78 (15):1625-1631. doi: 10.1007/s40265-018-0983-6.
Minor corrections:
(1) Abstract: The abstract must be rewritten to better explain the highlights of the review. Why are NA-based therapies more universal? What you intend to say with that statement? Choose a more appropriate word rather than "powerful" for delivery systems. Why those formulation conditions are important? That is not clear for the reader.
We have considered comment and have restructured the abstract.
(2) Keywords: cellular trafficking is not a representative word of the review, please change for another one.
Page 1 line 33: Keywords were changed: Cell-penetrating peptide, nanoparticle, nucleic acid, delivery, self-assembly
(3) Explain why peptide-based vectors are inexpensive for the delivery of therapeutic NAs? (lines 90-91, page 2).
Page 2 lines 94-95: We agree with Reviewer 1 that the sentence was not clear and changed the sentence in “Moreover, peptide-based vectors are now considered as suitable candidates for the delivery of therapeutic NAs due to their easy automated synthesis, single-step formulation and biocompatible properties.”
(4) Please change PEG moieties to PEGylated in Figure 1 caption;
We have done the change required by reviewer 1.
(5) Please rewrite the first sentence of Poly-cationic family topic (lines 123-124) and better introduce that topic.
We have changed the first sentence of the poly-cationic family section:
Page 4 lines 143-146: “Peptides made only with lysine and arginine, named poly-lysine (Poly-Lys) and poly-arginine (Poly-Arg), are one of the first artificial CPPs which were evaluated for their internalization in living cells. Prof. S. Futaki’s group was one of the pioneers working on the effect of the positive charge on cell transfection poly-cationic peptides with 4 to 16 residues [29].”
(6) PEGylation section: There are other advantages regarding PEGylation of PBNs, please explain them in the beginning of the topic. Three important works regarding PEGylation of CPPs should be cited in the review: doi: 10.2174/1568026620666200128142603 (general improvements through PEGylation for CPPs), dx.doi.org/10.1590/s2175-97902018000001009 (general improvements through PEGylation for biobetters) and doi.org/10.1039/C9TB00590K (thermostability enhancement)
We agree with Reviewer 1 that our section about PEGylation lacks explanations about the advantages of shielding strategies and use of PEG. We thus include a more detailed paragraph in the section:
Page 6 lines 422-429: “One major drawback of the PBNs as in vivo delivery system is their short life span in the blood circulation. Their size and their charge could influence the recognition by specific defense systems of the body then the absorption by the system of mononuclear phagocytes, which prevents them from entering other tissues. To circumvent this limitation, PEGylation has been considered as a significant shielding strategy. Indeed PEGylation of nanoparticles have several pharmacological advantages such as an improved drug solu-bility, an increased drug stability and an extended circulating life [108]. Moreover a reduced toxicity and rate of kidney clearance, an enhanced protection from proteolytic degradation, a decreased immunogenicity and a minimal loss of biological activity might be also noticed when nanoparticles are PEGylated.”
We thus inserted the following reference which is more focused on nanoparticles than proteins: Gulati NM, Stewart PL, Steinmetz NF. Bioinspired Shielding Strategies for Nanoparticle Drug Delivery Applications. Mol Pharm. 2018 Aug 6;15(8):2900-2909. doi: 10.1021/acs.molpharmaceut.8b00292.
With regard to the other references proposed by Reviewer 1, we did the choice to cite a review more focused on PEGylation of nanoparticles (which is closer to the field of PBN) than a review dealing with PEGylation of proteins or peptides.

Reviewer 2 Report
This is a comprehensive review mentioning different peptides and their specific applications for nucleic acid delivery. I think the manuscript is well organized but there are some minor points that could be addressed;
- Please add an outline to direct the readers into different sections at the beginning of the manuscript
- Adding some illustrations citing specific studies mentioned in the manuscript would help drawing reader's attention
- There are minor spelling and wording that needs some improvement
- It would be good to mention/justify the advantages of peptides over polymers, inorganic nanoparticles and liposomes, which are the mostly used materials for non-viral nucleic acid delivery
- The positive charge sometimes may cause toxicity and it may be good to provide a better justification for the non-toxicity of these positively charged peptides or the approached to make them non-toxic
- Peptide can also be used in conjugation with polymers and other type of materials to design efficient delivery systems. So a separate section discussion the synergistic use of peptides with other materials for delivery purposes would be appropriate.
- Looks like this review is more focused on CPPs but fusogenic peptides are also considered as promising alternatives. It may also be valuable to add a short section on fusogenic peptides or give some examples about their potential uses.
Author Response
Please also see the attachment
Review Report Form
Open Review REVIEWER 2
Comments and Suggestions for Authors
This is a comprehensive review mentioning different peptides and their specific applications for nucleic acid delivery. I think the manuscript is well organized but there are some minor points that could be addressed;
- Please add an outline to direct the readers into different sections at the beginning of the manuscript
We thank the reviewer 2 for this comment. In order to clarify the focus of the review, we have changed the end of the introduction:
Page 4 line 121-133 : “In this Review, we present different peptide families used for PBN formulation in the presence of various NAs (Table 1) which were used and published during the period between 2015 and 2021 (even if their design was reported years previously). Especially, we have focused this review on peptide-based nanoparticles formed by self-assembly of peptides which are essentially native or modified CPPs (PEGylated, grafted to fatty acids or fusogenic moieties, etc.). CPPs used in the covalent strategy or to decorate other nanoparticles such as polymers, nanotubes or even virus are reported elsewhere [29,30].
In details, we have first summarized seven “main” CPP families such as the poly-cationic, GALA/KALA/RALA, PepFect/NickFect, CADY-K/RICK, WRAP, C6, Mpge families and secondly highlighted new developed CPPs with high potential (see section “Other CPPs forming PBNs”). Finally, we have recapitulated known PBN optimization such as PEGylation and different targeting strategies (Figure 2) which are important for the development of “intelligent” PBNs in view of a pharmacological application.”
- Adding some illustrations citing specific studies mentioned in the manuscript would help drawing reader's attention
We have added a second figure (Figure 2, see below) illustrating the advantages and disadvantages of the PBNs modifications (PEGylation, Acylation, Targeting).
However, we decided to illustrate this review only with schemes because:
- Based on the multitude of PBNs applications, it is not easy to privilege one over the others.
- Figures showing published results from other authors needs publication agreements with the corresponding fees.
- There are minor spelling and wording that needs some improvement
We have considered comments of Reviewer 2 by improving the English language through the whole manuscript. All changes were noticed in red in the manuscript.
- It would be good to mention/justify the advantages of peptides over polymers, inorganic nanoparticles and liposomes, which are the mostly used materials for non-viral nucleic acid delivery
We thanked the reviewer for this suggestion. However, the field of nanomedicine covers a lot of delivery systems used in different physio-pathological context making a direct comparison difficult. Furthermore, such comparison will be beyond the scope of this review.
- The positive charge sometimes may cause toxicity and it may be good to provide a better justification for the non-toxicity of these positively charged peptides or the approached to make them non-toxic
We agreed with reviewer that positive charges are often toxic. One possibility to reduce surface charges is to graft PEG entities on the nanoparticle. The PEGylation shielded the nanoparticle surface charges making them more invisible and therefore less toxic.
- Peptide can also be used in conjugation with polymers and other type of materials to design efficient delivery systems. So a separate section discussion the synergistic use of peptides with other materials for delivery purposes would be appropriate.
We agreed with the reviewer’s comment. However, this review is only focalized on CPP able to self-assemble in the presence of nucleic acids in order to form peptide-based nanoparticles.
We have added in the introduction the following sentence:
Page 4 line 125-127: “CPPs used in the covalent strategy or to decorate other nanoparticles such as polymers, nanotubes or even virus are reported elsewhere [29,30].”
- Looks like this review is more focused on CPPs but fusogenic peptides are also considered as promising alternatives. It may also be valuable to add a short section on fusogenic peptides or give some examples about their potential uses.
We agree with the reviewer 2 that fusogenic peptides are also considered as promising alternatives. In our review, we chose to only focus on peptides that are able to self-assemble in peptide-based nanoparticles, whatever their nature, origins or features. As the non-covalent peptide-nucleic acids strategy was first applied with some CPPs (MPG, Transportan), it was relatively normal that most of peptides developed in the peptide-based nanoparticles approach belonged then to this CPPs class. However the only common requirement we took into account in our paper was the ability to self-assemble in nanoparticles with nucleic acids by themselves. Moreover we decided to present these peptides by family (names, groups) instead of a classification by properties such as charged, amphipathic, fusogenic or antimicrobial nature (as Francesca Milletti did in her review Drug Discov Today 2012), for two reasons: (1) the frontier between classes is sometimes not so clear (ex CPPs vs antimicrobial peptides vs fusogenic peptides) (2) the self-assembly ability is the main property driving peptide-based nanoparticle formation. Finally, even if we did not display a specific section about fusogenic peptides, they are still present as shown in the “3. GALA/KALA/RALA family” for the GALA peptide.

Round 2
Reviewer 1 Report
The authors addressed the main concerns from the required reviews (both major and minor critics), the revised version of the manuscript appears to be suitable for publication at Biomedicines. I congratulate the authors for their dedicated work in revising the article, especially with the inclusion of Figure 2, which raised the quality of the review. Therefore, I recommend the publication of the aforementioned article.